# 3D Reverse-Time Migration Imaging for Multiple Cross-Hole Research and Multiple Sensor Settings of Cross-Hole Seismic Exploration

**DOI:** 10.3390/s24030815

**Published:** 2024-01-26

**Authors:** Fei Cheng, Daicheng Peng, Sansheng Yang

**Affiliations:** 1Hubei Key Laboratory of Marine Geological Resources, China University of Geosciences, Wuhan 430074, China; chengfly03@126.com; 2Key Laboratory of Exploration Technologies for Oil and Gas Resource, Ministry of Education, Yangtze University, Wuhan 430100, China; 3China State Shipbuilding Corporation 722 Research Institute, Wuhan 430205, China; yss_200808@163.com

**Keywords:** three-dimensional imaging, cross-hole seismic exploration, reverse-time migration, multiple cross-holes, multiple sensor settings

## Abstract

The two-dimensional (2D) cross-hole seismic computed tomography (CT) imaging acquisition method has the potential to characterize the target zone optimally compared to surface seismic surveys. It has wide applications in oil and gas exploration, engineering geology, etc. Limited to 2D hole velocity profiling, this method cannot acquire three-dimensional (3D) information on lateral geological structures outside the profile. Additionally, the sensor data received by cross-hole seismic exploration constitute responses from geological bodies in 3D space and are potentially affected by objects outside the well profiles, distorting the imaging results and geological interpretation. This paper proposes a 3D cross-hole acoustic wave reverse-time migration imaging method to capture 3D cross-hole geological structures using sensor settings in multi-cross-hole seismic research. Based on the analysis of resulting 3D cross-hole images under varying sensor settings, optimizing the observation system can aid in the cost-efficient obtainment of the 3D underground structure distribution. To verify this method’s effectiveness on 3D cross-hole structure imaging, numerical simulations were conducted on four typical geological models regarding layers, local high-velocity zones, large dip angles, and faults. The results verify the model’s superiority in providing more reliable and accurate 3D geological information for cross-hole seismic exploration, presenting a theoretical basis for processing and interpreting cross-hole data.

## 1. Introduction

The cross-hole seismic acquisition method has the potential to capture the target zone precisely, compared to surface reflection or refraction surveys. Surface seismic methods are suitable for near-surface conditions; however, they face challenges in transmitting sufficient energy through the weathered zone to desired depths due to the strong attenuation of seismic waves in unsaturated, unconsolidated sediments [1,2,3]. The cross-hole seismic imaging method has gained wide utilization in geotechnical engineering parameter calculation [4], tunnel and cavity detection [5,6,7], hydrogeology surveys [8,9,10], rock and aquifer distribution detection [11,12,13], reservoir descriptions [14,15,16], rock fragmentation descriptions [17], civil engineering [18], detection of hydrocarbon reservoirs [19,20,21] and so on. Despite this, this method also has deficiencies; for example, when cross-hole seismic receivers are arranged near the target body, the wavefield in profile between two wells is greatly influenced due to the medium inhomogeneity (especially the high-velocity target) in 3D space, leading to image distortion [22]. Additionally, 2D travel time imaging is limited to the two adjacent wells instead of multi-well distribution in 3D space, resulting in limited spatial information [23]. The multi-well profile is applied to interpret the 3D velocity structure between wells [24]. This processing method typically inverses the 2D profile between two adjacent wells independently, followed by interpolation or extension to obtain the 3D velocity (slowness) structure. The 3D model construction method, which extrapolates the connecting well profiles using 2D profiles for interpolation and least squares fitting, can improve the accuracy of the profile and its adjacent structure, yet this is insufficient to obtain accurate 3D structures [25].

In order to better obtain an accurate description of a 3D geological structure between wells, scholars have carried out 3D imaging works in recent years. Among these, 3D cross-hole seismic reflection wave imaging is widely used and has achieved good results [26]. There are two main imaging methods used in 3D cross-hole seismic reflection wave imaging: ray theory and wave equation theory [27]. The seismic reflection wave imaging basis of ray theory can provide media images quickly and effortlessly. However, it cannot perform high-precision imaging of complex structures, and it cannot fully utilize 3D wavefield information for imaging [28]. The reverse-time migration (RTM) method based on the wave equation can adapt to strong lateral variations in the velocity field and is considered one of the most accurate imaging methods [29,30,31]. In recent years, the 3D RTM method has been applied in the field with promising cross-hole results [32,33,34]. Compared with traditional ray-based cross-well methods, RTM includes more wavefield information, allowing seismic wave energy to be returned to its true location in space, obtaining more accurate geological structures and providing higher-quality images [35,36]. In terms of cross-hole seismic RTM, researchers have proposed 2D cross-hole seismic transverse RTM imaging of acoustic waves and elastic waves in isotropic media [37]. Subsequently, to provide richer and more effective information for a better understanding of the geological characteristics between wells, the researchers derived the first-order velocity-stress elastic wave equation of longitudinal and transverse wave decoupling based on an inhomogeneous medium and obtained 2D cross-hole longitudinal and shear wave imaging sections [38]. However, current research is limited to 2D cross-hole RTM imaging and does not utilize information from multiple wells, so it is impossible to obtain the entire 3D geological structure [39,40].

To solve the limitations of conventional 2D cross-hole imaging and enhance the precision of 3D cross-hole exploration, we propose a 3D cross-hole acoustic wave RTM im-aging method and sensor settings in multi-cross-hole seismic research and obtained the 3D cross-hole first-arrival travel time field using the fast marching method (FMM) algorithm. Then, the observation data of multiple wells and the excitation time imaging conditions were employed to test the 3D cross-hole acoustic RTM imaging method of multi-well data and multi-cross-hole sensor settings, thus capturing the 3D geological structure. To verify the effectiveness of this method, numerical simulations were conducted on layered, local high-velocity, large dip angle, and fault models. The results demonstrate that 3D imaging is advantageous, providing more reliable and accurate 3D geological information for cross-hole seismic exploration and a theoretical basis for processing and interpreting cross-hole data. Furthermore, analysis of the 3D cross-hole imaging results obtained under various sensor settings shows that the design of the observation system can be improved to acquire the cost-efficient distribution of 3D underground structures.

## 2. Methodology

We developed an acoustic wave RTM based on an acoustic two-way wave equation using fused deposition modeling for multi-cross-hole imaging in a 3D tunnel. The following sections present the 3D acoustic two-way wave equations, imaging conditions, seismic wave travel time computation, and reduction of RTM artifacts.

### 2.1. 3D Acoustic Two-Way Wave Equation

In the constant density medium, the 3D acoustic two-way wave equation is written as follows [41]:(1)$$\frac{1}{v^{2}}\frac{\partial^{2}p}{\partial t^{2}}=\frac{\partial^{2}p}{\partial x^{2}}+\frac{\partial^{2}p}{\partial y^{2}}+\frac{\partial^{2}p}{\partial z^{2}}+s$$
where *p* = *p*(*x*, *y*, *z*, *t*) denotes the pressure field, *v* = *v*(*x*, *y*, *z*) represents the velocity, and *s* = *s*(*x*, *y*, *z*, *t*) refers to the source term. To realize the backpropagation of the wave equation, a suitable wave propagator is typically constructed by the finite difference method in RTM, which discretizes the second-order partial derivative in Formula (1) and replaces the differential equation with the difference one. Subsequently, wavefield extrapolation is achieved. In this technique, we adopt convolutional perfectly matched layer boundary conditions to suppress and absorb the boundary wavefield.

### 2.2. Imaging Condition

Time-reversed extrapolation is performed along the timeline using the RTM method based on the wave equation, primarily including (i) forward propagation of the source wavefields, (ii) extrapolation of the time-reversed scattered wavefield measured at receiver positions, and (iii) imaging using imaging conditions. The pre-stack RTM imaging scheme can be characterized by several imaging conditions. In this paper, we used the following excitation time imaging condition [42,43]:(2)$$\left\{ \begin{array}{l} Image\left(x,y,z\right)=\sum_{t}r\left(x,y,z,t\right)f\left(x,y,z,t\right)\\ f\left(x,y,z,t\right)=\left\{ \begin{array}{cc} 1 & x=x^{\prime },y=y^{\prime },z=z^{\prime },t=t_{d}\left(x^{\prime },y^{\prime },z^{\prime }\right)\\ 0 & other \end{array}\right. \end{array}\right.$$
where *t* is the maximum time of the wavefield records. The wavefield values can be extracted from the reverse-time wavefield *r*(*x*, *y*, *z*, *t*) extrapolation according to the excitation time *t_d_*(*x*’, *y*’, *z*’), and the migration section image (*x*, *y*, *z*) can be obtained after the summation of all the extracted wavefield values.

### 2.3. Computation of Seismic Wave Travel Time

In calculating imaging time, the FMM algorithm by Sethian is adopted [44,45]. This method realizes a fast calculation along the minimum time wavefront; additionally, it compares and updates the sample on the narrow band produced by grid expansion in line with the time calculated in the windward area and the leeward area. This method is considered unconditionally stable and can obtain an accurate solution despite a dramatic change in the model speed. According to Huygens’ principle, the Eikonal equation (the time differential equation) used to calculate the travel time in 3D space is expressed as follows:(3)|∇T(x,y,z)|=S(x,y,z)

In Equation (3), (*x*, *y*, *z*) is the spatial coordinate, and *T* and *S* are the travel time and slowness distribution functions of the underground medium, respectively. The FMM algorithm mainly uses the upwind difference scheme to solve the Eikonal equation. For a discrete grid node (*i*, *j*, *k*), the time gradient items of the Eikonal equation can be expressed in an upwind difference scheme as follows:(4)$$|\nabla T|={\left[\begin{array}{l} \max{\left(D_{n}^{-x}T_{i,j,k},0\right)}^{2}+\min{\left(D_{n}^{+x}T_{i,j,k},0\right)}^{2}\\ +\max{\left(D_{n}^{-y}T_{i,j,k},0\right)}^{2}+\min{\left(D_{n}^{+y}T_{i,j,k},0\right)}^{2}\\ +\max{\left(D_{n}^{-z}T_{i,j,k},0\right)}^{2}+\min{\left(D_{n}^{+z}T_{i,j,k},0\right)}^{2} \end{array}\right]}^{1/2}=S_{i,j,k}$$
where the value $T_{i,j,k}$ is the travel time of the grid nodes (*i*, *j*, *k*), and $S_{i,j,k}$ is the slowness. $D_{n}^{-x}T_{i,j,k}$ and $D_{n}^{+x}T_{i,j,k}$ are, respectively, the n-order forward and backward difference operator on the x-direction. $D_{n}^{-y}T_{i,j,k}$ and $D_{n}^{+y}T_{i,j,k}$ are on the *y*-direction; $D_{n}^{-z}T_{i,j,k}$ and $D_{n}^{+z}T_{i,j,k}$ are on the z-direction. The max and min in the formula mean the maximum and minimum of the three values. The stability of this fast advance algorithm can realize the cross-hole travel time calculation with complex media well and guarantee the quality of the imaging profile.

### 2.4. RTM Artifacts

The RTM can adapt to notable lateral variations in the velocity field, yet the application of imaging conditions can cause low-frequency artifacts [46,47]. The Poynting vector and Laplace filtering are deployed in this article to denoise the RTM images. The Poynting vector can be expressed as:(5)$$\left\{ \begin{array}{l} d_{x}=dp/dx\cdot sign\left(dp/dt\right)\\ d_{y}=dp/dy\cdot sign\left(dp/dt\right)\\ d_{z}=dp/dz\cdot sign\left(dp/dt\right) \end{array}\right.$$
where *p* denotes the pressure field, $dp/d_{i}\left(i=x,y,z\right)$ represents the displacement vector of the particle, and $d_{i}\left(i=x,y,z\right)$ denotes a vector in the direction of the ray. When d*i* is positive, the wavefield propagates along the positive direction of the *i*-axis, whereas when d*i* is negative, the wavefield propagates along the negative direction of the *i*-axis. Therefore, the corresponding filtering factor can be designed to denoise the wavefield according to Equation (5). Then, we can use Laplace filtering to attenuate the artifacts in the 3D-imaged data body. The Laplace operator can be expressed as:(6)$$\nabla^{2}=\frac{\partial^{2}}{\partial x^{2}}+\frac{\partial^{2}}{\partial y^{2}}+\frac{\partial^{2}}{\partial z^{2}}=-\frac{4\omega^{2}\cos^{2}\theta }{v^{2}}$$
where $\nabla^{2}$ is the Laplace operator, α is the angle of incidence, ω denotes the frequency, and v denotes the medium velocity. With this filter factor, noise in imaging will be entirely attenuated where the incident angle is near 90° and suppressed in parts of noise in imaging with incident angles below 90°.

### 2.5. Implementations

Figure 1 shows the implementation flowchart of the proposed method. Initially, a 3D modeling of multiple cross-holes was established, and reasonable absorbing boundary conditions were set up. On this basis, we performed an extrapolation simulation of 3D acoustic wavefield and cross-hole RTM imaging. The RTM with the excitation time imaging condition consists of three steps: (1) computation of the imaging condition using FMM; (2) acquisition of the extrapolated recorder wavefield data from T = t_max_ to T = 0; (3) application of the excitation time imaging condition. Following this, the Poynting vector and Laplace filtering are used to attenuate the artifacts in 3D RTM images. Based on the analysis of resulting 3D cross-hole images obtained under varying sensor settings, the design of the observation system can be improved, and the distribution of 3D underground structures can be obtained at an affordable cost.

## 3. Survey Layout and Modeling

### 3.1. Survey Layout

The cross-hole seismic method involves the explosion of seismic sources and seismic signal reception within wells. Its observation system is different from that of the 3D seismic method on the ground, and the survey line can be arranged on a large scale and at high density. As shown in Figure 2, a square exploration area measuring 45.0 m × 45.0 m is selected as an example, and 12 wells are arranged for cross-hole 3D RTM imaging re-search. The simulation grid spacing for imaging is dx = dy = dz = 0.5 m. The coordinates of the wells in the XOY plane are shown in Table 1. The source is first exploded in well-1, and the rest of the wells are used to receive the seismic signals. The sources are located in well-1 with a spacing of 1.0 mm, and sensors are arranged in well-3 and well-11 with the same spacing as the sources. Each line has 100 sensors in the well. Then, wells 2–12 are sequentially used as exploding wells for data acquisition. An explosion source is assumed in the acoustic wavefield extrapolation. The wavelet source is a Ricker wavelet with a delay of 3.0 ms and a frequency of 120 Hz.

### 3.2. Design of the Theoretical Model

Two-dimensional cross-hole computed tomography (CT) imaging does not utilize information from multiple wells, resulting in a low utilization rate of spatial well locations. In addition, it also possesses problems related to poor lateral continuity of images and the inability to image large dip angle interfaces. Four types of theoretical geological models were designed to illustrate the effectiveness of the multi-well 3D RTM imaging method on local high-speed target bodies and large dip angle interfaces as well as its improved continuity in the lateral direction. The 3D models measure 45.0 m × 45.0 m × 100.0 m, as shown in Figure 3.

To validate the developed RTM imaging method, we established a geological model with three horizontal strata (referred to as model-1), as shown in Figure 3, and the two interfaces are 30 m and 80 m in depth, respectively. Model-2 is a layered structure embedded with a high-velocity ellipsoid to analyze the method’s effectiveness for imaging local high-velocity models. The ellipsoid radii are a = 5.0 m, b = 5.0 m, and c = 10.0 m, and the center is located at 22.5 m, 22.5 m, and 22.5 m, respectively. Model-3, with a 60° dip angle, is designed to illustrate the effectiveness of interface imaging with large dip angles. The inclined interface is parallel to the Y axis, the X and Z coordinates of the intersection line with the first interface are 35.0 m and 30.0 m, respectively, and the depths of the other two horizontal interfaces are 30.0 m and 75.0 m, respectively. Model-4 concerns geological faults and is designed to study the imaging quality for complicated geological conditions. The top and bottom interface depths of the hanging wall are 45.0 m and 75.0 m, respectively; the counterparts of the footwall are 30.0 m and 60.0 m, respectively. The fault plane is parallel to the Y axis, and its dip angle is 60°. The X and Z coordinates of the intersection line with the top footwell interface are 10.0 m and 30.0 m, respectively. Table 2 provides a detailed description of each model’s velocity parameters.

## 4. Verifications and Discussion

### 4.1. RTM Imaging Artifact Attenuation

The application of imaging conditions can attenuate the frequency artifacts that interfere with effective information. These artifacts affect the 3D imaging results and structural interpretation. For this reason, the Poynting vector and Laplace filtering were employed to obtain the final 3D imaging based on the initial photographing. To validate the current attenuation method of RTM imaging artifacts, we designed a layered model (Figure 4a). In the first layer, the P-wave velocity is 2000.0 m/s, and the density is 1500.0 kg/m^3^; in the next layer, the P-wave velocity is increased to 2400.0 m/s, and the density is elevated to 2000.0 kg/m^3^. The unprocessed RTM imaging is shown in Figure 4b. It can be seen that the RTM imaging results are distorted by low-frequency artifacts caused by reflected waves, which interfere with the imaging layer interfaces. As shown in Figure 4c, the reflected wave noise in the obtained images can be eliminated and has no effects on the interface imaging when using the Poynting vector to attenuate the reflected wave noise during the migration, and some artifacts remain. Figure 4d shows the imaging results using the Poynting vector and Laplace filtering. A high signal-to-noise ratio is observed with the attenuation of low-frequency artifacts.

### 4.2. Discussion of Observation System with Multiple Sensor Settings

Considering actual needs and costs, the sensors and survey lines of cross-hole seismic surveys cannot be arbitrarily arranged akin to those of the surface seismic. For this reason, an optimized observation system with sensor settings that provides reliable 3D cross-hole structures is necessary for obtaining the distribution of underground structures at minimal cost. The observation system with 16 well locations for RTM imaging is shown in Figure 5a. Figure 5b presents the experimental layered model. The specific data collection methods are as follows: The source was first exploded in well-1, and the seismic signals were received at the other 11 wells. Then, the wells numbered between two and sixteen were used as exploding wells for data acquisition. To obtain the distribution of under-ground structures in a cost-efficient manner, we designed various wells and sensor set-tings with different numbers and locations to discuss the varying influences of the observation system with different sensor settings on the 3D RTM imaging results. According to Figure 6, six types of wells and sensor settings for various observation systems are presented.

Figure 6a schematically displays the observation system at four well locations, and the 3D imaging results are shown in Figure 7a. It can be seen that the imaging area is small, and a relatively strong event is detected only in the center area. By contrast, the event is discontinuous in other regions, impeding the analysis and interpretation of the local geological structure. The imaging results under the six-well observation system (Figure 6b) are illustrated in Figure 7b. Despite the small area of strong energy, the energy shape is slightly enhanced compared to the four wells. Local discontinuities can be found in the event axis, potentially impacting the interpretation of results. Figure 6c–e shows different observation systems with eight well locations. It can be seen from Figure 6c that due to the missing well locations in the four corners, the imaging results (Figure 7c) exhibit corresponding events disappearing in these areas. However, the imaging results in the octagonal areas are favorable, and the resulting events are continuous, properly reflecting the interface information. In this sense, this observation system is suitable for obtaining the geological information of intermediate areas. Figure 7d shows the imaging results obtained under the observation system in Figure 6d. Although the event energy in the middle interface is weaker than in Figure 7c, the imaging results can cover the entire interface. The energy at 7.0–12.0 m and 44.0–49.0 m in the Y direction is weaker than in the other direction, and the events exhibit the same positive phase. The optimal imaging effects are concentrated in the rectangular area in the interfacial center and near the wells under this observation system. Figure 7e presents the imaging results obtained under the observation system in Figure 6e. Similarly, these results cover the entire interface, and the events re-main in a consistent phase despite having lower energy than that in Figure 7d. Figure 7f provides the imaging results of the observation system (Figure 6f) at 12 well locations. The whole interface is favorably imaged, and the strong energy events correspond to the interface position, indicating the accuracy of the method. The fully characterized interface demonstrates that this 3D cross-hole RTM imaging method is effective and feasible when performed with an observation system with 12 well locations.

## 5. Multiple Cross-Hole 3D RTM Image Results and Analysis

### 5.1. Model-1

In the case of Model-1, the seismic signals received in well-3 and well-7 when well-1 exploded at 15 m depth are illustrated in Figure 8a,b. The cross-hole wavefield is more complicated than the ground records. For a simple three-layer model, the seismic records comprise direct, transmitted, reflected, interlayer multiple reflection and transmission waves. In addition, the refracted wave caused by the higher velocity of the lower layer can also be seen in Figure 8b. The wavefield recording is accomplished and can correctly reflect the wavefield characteristics of the 3D cross-hole environment.

Figure 8c displays the pre-stack RTM imaging section of Model-1, and the spatial sampling interval is 1 m. Limited by the actual situation, the observation system of the cross-hole seismic method cannot be arranged in the same way as that of surface seismic imaging, interrupting the continuity of the imaging section. However, the wavefield energy is shown to be focused on the real interface, which is consistent with the actual model, demonstrating that the imaging results have good lateral continuity. It is indicated that the proposed imaging method can obtain the 3D cross-hole geological structure, and the results are accurate and reliable.

### 5.2. Model-2

For Model-2, the seismic signals received from well-3 and well-7 when subjected to the explosion in well-1 at 15 m depth are shown in Figure 9a,b. From Figure 9a, in addition to the waves from the interface, the seismic records also include the diffracted spherical waves. Given that well-7 and the shot well divide the high-velocity sphere, the diffracted waves from the sphere and the direct waves overlap in Figure 9b. Due to the high-velocity sphere, the type of the first arrival wavefield has been changed to a black arrow in the figure. After removing the direct waves, the 3D cross-hole imaging results of the local high-velocity model-2 were obtained using 3D RTM based on the acoustic equation, as shown in Figure 9c. Accordingly, the wavefield energy is mainly distributed on the two interfaces and the sphere, and the interface locations coincide with those of the events. The imaging radius of the spheroid in the short axis is slightly smaller than in the real model, and this difference does not affect the understanding or judgment of the abnormality, meaning that the imaging results are still in good agreement with the real model. Based on the above analysis, the 3D cross-hole RTM imaging method based on multiple wells exhibits advantages in capturing the geological structure in 3D space. In addition to the model with horizontal interfaces, the local heterogeneous model also demonstrates good results. In this sense, the proposed method can obtain the geological information of the lateral medium, accurately depict the shape of the local heterogeneous body, and accurately obtain the 3D geological structure, effectively overcoming the inability to image lateral media in 2D conditions. To summarize, the 3D cross-hole RTM imaging method based on multiple wells can better solve the problem of “boulder” detection in engineering exploration and provide a new way for engineering exploration.

### 5.3. Model-3

Figure 10a presents the 3D cross-hole imaging results of the model with a large dip angle interface. The wavefield energy mainly concentrates on the three interfaces, and the event location in the image coincides with the real model, indicating that the imaging results are correct and reliable. In addition, the event energies of the upper and lower interfaces are stronger than those of the interface with a large dip angle. In the intersection of the inclined and the lower interfaces, the energy is weakened due to the relatively low superposition in this area. This phenomenon typically does not affect the analysis or interpretation of the whole structure. In summary, the 3D cross-hole multi-well RTM imaging method can obtain the cross-hole 3D structure and overcome the defects of CT imaging in large dip angle interface imaging. In other words, the method can obtain geological structures in 3D regions, presenting a basis for interpreting cross-hole 3D structures.

### 5.4. Model-4

Figure 10b summarizes the 3D cross-hole imaging results of a complicated model with faults. For this 3D model, the upper and bottom interfaces of the hanging wall exhibit strong event energies, and those of the footwall have continuous events. It is indicated that this method can depict the upper and bottom interfaces of the hanging wall and footwall as well as the fault plane. The imaged event locations are consistent with the actual model, verifying the validity and reliability of the imaging results. Based on the above analysis, the 3D cross-hole RTM imaging method based on multiple wells can also be used for complicated 3D fault models with accurate and reliable outcomes.

### 5.5. Imaging Results of Data Contaminated with Random Noise

The above model imaging is based on noise-free data. There will inevitably be a certain degree of noise in the actual collected data. To test the practicability of the imaging method proposed in this article, data contaminated with random noise are added to the numerical simulation of the theoretical model, and then the imaging results are analyzed. Model-1 is selected for validation. Twenty percent Gaussian random white noise is added to the obtained simulation records, and the single-shot simulation recording of data contaminated with random noise is shown in Figure 11a. It can be seen from the record that the noise is randomly distributed in the record, in which the effective reflected wave information is no longer as clear as in the noiseless condition, and only the event axis can be vaguely seen, which is consistent with the actual recording situation. The 3D RTM imaging results of data contaminated with random noise are shown in Figure 11b. From the imaging results, it can be found that after adding 20% noise, the wavefield energy in the imaging results can also be concentrated at the location of the real interface, which can also match the real model well. There is a certain amount of noise present relative to the noise-free imaging results, but the noise is relatively weak and does not affect the interpretation of the imaging results. This shows that the 3D RTM imaging method for multiple cross-hole and multiple sensor settings proposed in this paper has good noise immunity and can be applied well to practical cross-hole 3D imaging.

## 6. Conclusions

In this study, cross-hole wavefield simulation was achieved using an acoustic two-way wave equation with fused deposition modeling based on the designed 3D geological model. Then, multi-well observation data and excitation time imaging conditions were employed to explore the effectiveness of the cross-hole 3D acoustic RTM imaging method in obtaining the 3D geological structure. The 3D cross-hole imaging results under varying sensor settings demonstrate the potential to improve the observation system layout, allowing for the cost-efficient obtainment of the distribution of 3D underground structures. Based on the observation system consisting of 12 wells, 3D cross-hole RTM imaging of multiple wells was performed on four typical models regarding horizontal layers, local heterogeneous bodies, large dip angle interfaces, and faults. The results show that the developed approach can improve the reliability and accuracy of the obtained 3D geological information for cross-hole seismic exploration. In addition, this method can capture the lateral structure variations unavailable in 2D imaging and address the problems of CT imaging on interfaces with large dip angles. Furthermore, the accuracy and reliability of the proposed method were verified using numerical simulations. To conclude, our method can be deployed to obtain actual 3D cross-hole geological structures with multiple wells, offering a reference for 3D cross-hole seismic exploration and the interpretation of cross-hole data.

## Figures and Tables

**Figure 1 sensors-24-00815-f001:**
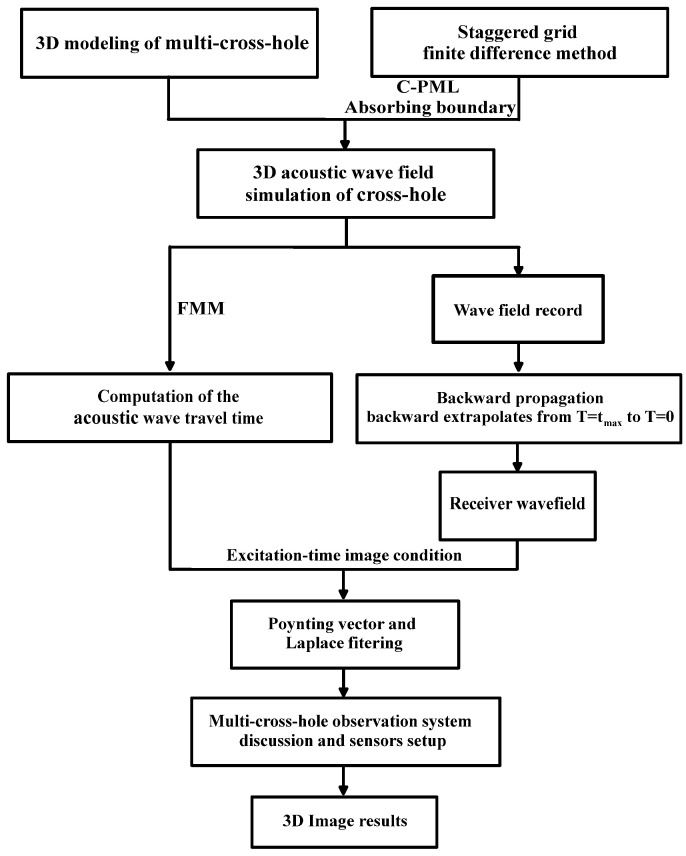
Flowchart of implementation.

**Figure 2 sensors-24-00815-f002:**
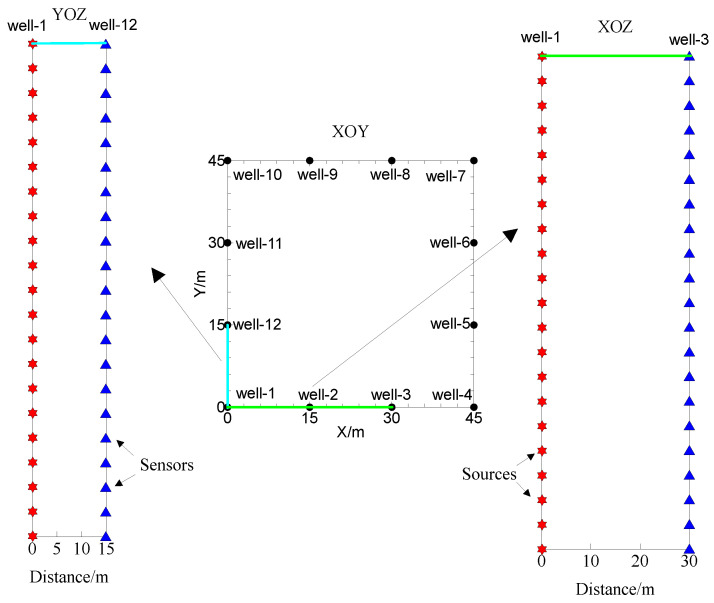
XOY plan of well location and the observation system with sensor settings of cross-hole seismic method.

**Figure 3 sensors-24-00815-f003:**
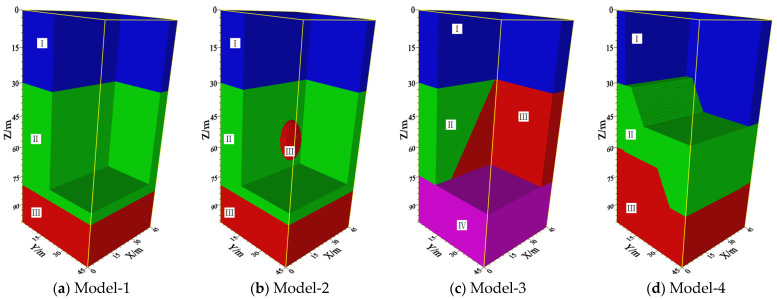
3D theoretical geological models: (**a**) three-horizontal-strata geological model; (**b**) layer structure embedded with a high-velocity ellipsoid model; (**c**) 60° dip angel geological model; (**d**) fault geological model. All parameters in the models are shown in Table 2.

**Figure 4 sensors-24-00815-f004:**
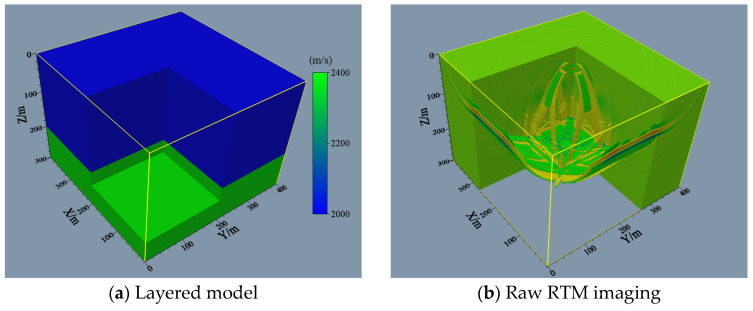
Comparison of single-shot RTM image denoising of 3D layered model.

**Figure 5 sensors-24-00815-f005:**
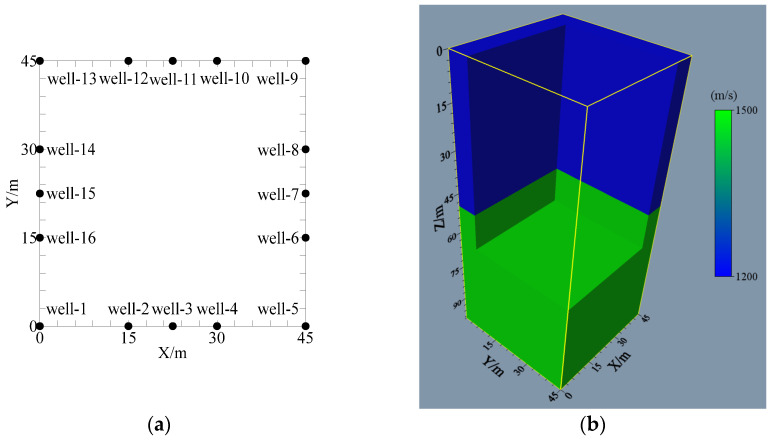
(**a**) Observation system of the wells’ location; (**b**) experimental layered model.

**Figure 6 sensors-24-00815-f006:**
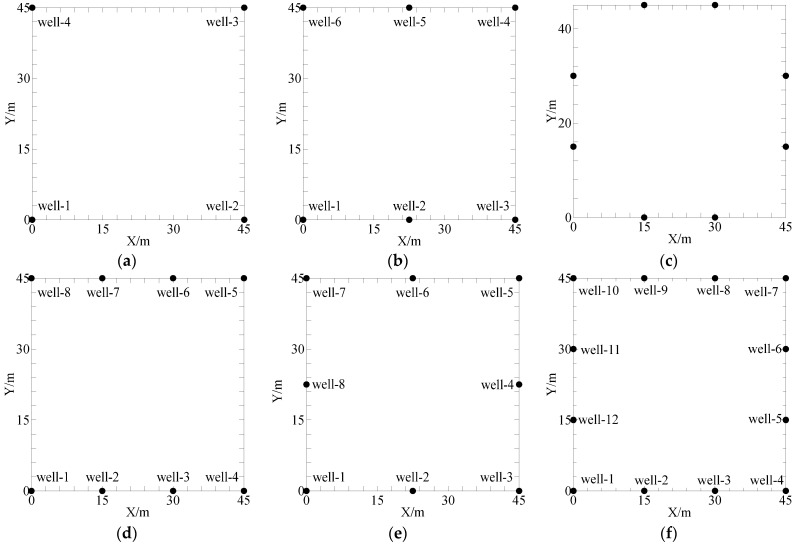
Schematic diagram of different well locations with sensor settings for different observation systems: (**a**) 4 wells; (**b**) 6 wells; (**c**–**e**) 8 wells; (**f**) 12 wells.

**Figure 7 sensors-24-00815-f007:**
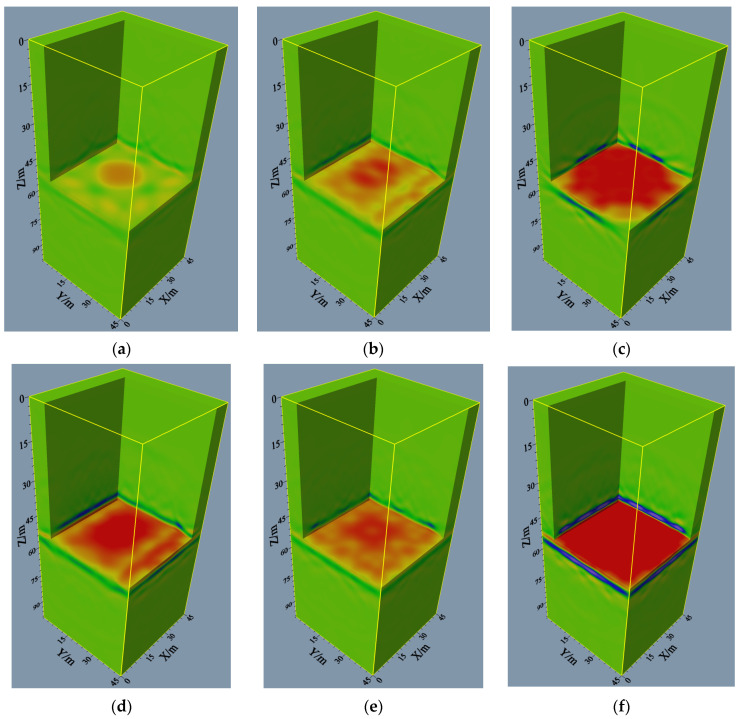
3D RTM imaging results for different well locations: (**a**) 4 wells; (**b**) 6 wells; (**c**–**e**) 8 wells; (**f**) 12 wells.

**Figure 8 sensors-24-00815-f008:**
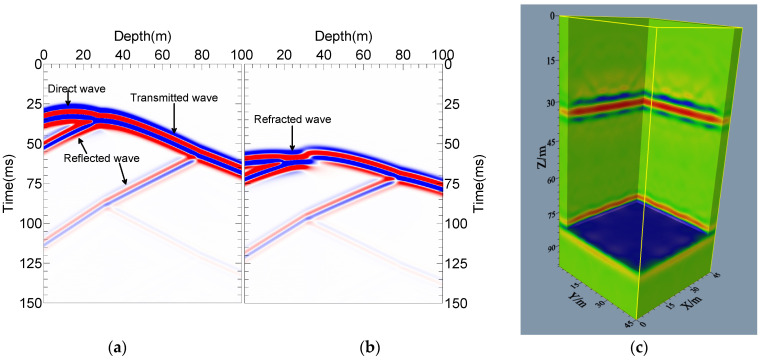
Model-1 records and 3D RTM imaging results: (**a**) well-3 received records; (**b**) well-7 received records; (**c**) 3D imaging results of model-1.

**Figure 9 sensors-24-00815-f009:**
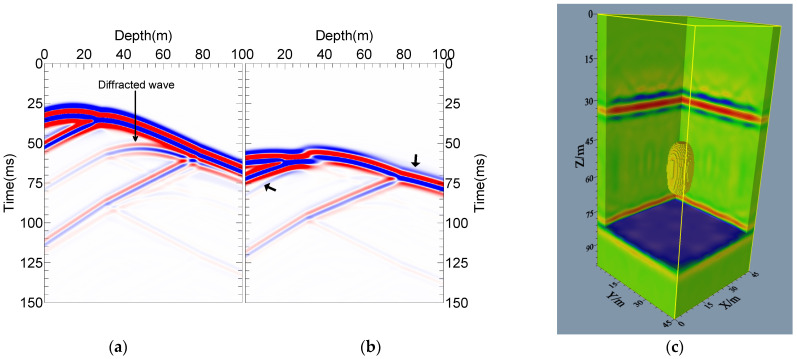
Model-2 records and 3D RTM imaging results: (**a**) well-3 received records; (**b**) well-7 received records; (**c**) 3D imaging results of model-2.

**Figure 10 sensors-24-00815-f010:**
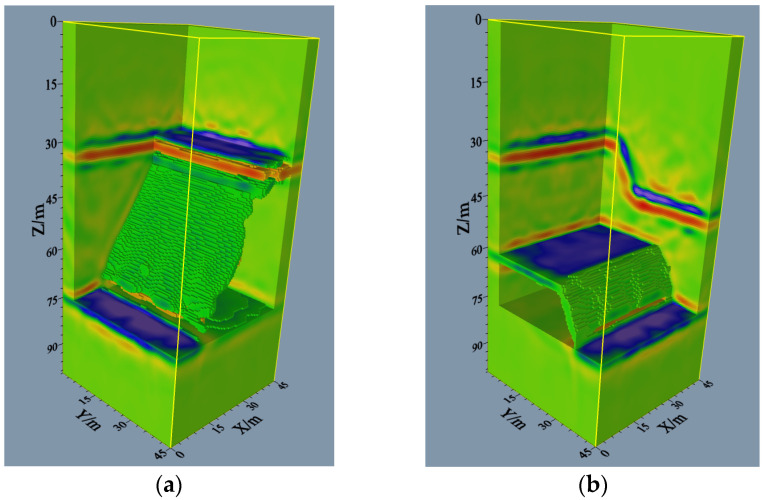
Model-3 and model-4 3D RTM imaging results: (**a**) imaging results of the model with large dip angle interface; (**b**) imaging result of a complicated model (fault model).

**Figure 11 sensors-24-00815-f011:**
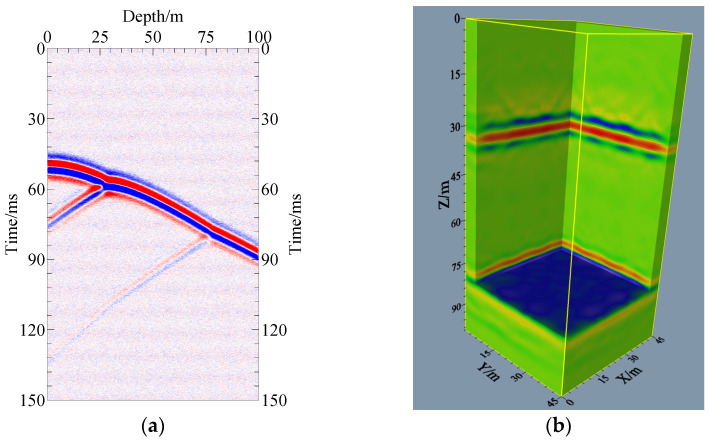
Model-1 records and 3D RTM imaging results of data contaminated with random noise: (**a**) single-shot recording of data contaminated with random noise; (**b**) imaging result of data contaminated with random noise.

**Table 1 sensors-24-00815-t001:** Well location layout in the XOY plane coordinates.

No.	Well-1	Well-2	Well-3	Well-4	Well-5	Well-6	Well-7	Well-8	Well-9	Well-10	Well-11	Well-12
X (m)	0.0	15.0	30.0	45.0	45.0	45.0	45.0	30.0	15.0	0.0	0.0	0.0
Y (m)	0.0	0.0	0.0	0.0	15.0	30.0	45.0	45.0	45.0	45.0	30.0	15.0

**Table 2 sensors-24-00815-t002:** Parameters of the models.

NO.	I	II	III	IV
*V_p_* (m/s)	1200.0	1500.0	1800.0	2000.0
*ρ* (kg/m^3^)	1800.0	2000.0	2200.0	2300.0

## Data Availability

The data presented in this study are available on request from the corresponding author.

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
