# Peer review of "3D Reverse-Time Migration Imaging for Multiple Cross-Hole Research and Multiple Sensor Settings of Cross-Hole Seismic Exploration"

_sensors, 2024, doi:10.3390/s24030815_

Round 1
Reviewer 1 Report
Comments and Suggestions for Authors
Dear Author:
Considering the manuscript with the manuscript ID: sensors-2838190, entitled “3D reverse time migration imaging for multiple cross-hole and multiple sensor settings of cross-hole seismic exploration", herewith I would like to submit my comments.
The manuscript is well prepared, the introduction needs to be improved, methodology needs to be explained better and results have to be illustrated in a better way. Although the manuscript technically sounds, but more information could be added to make it more attractive for readers. So, based on my review, I propose the manuscript to be subjected into moderate revision.
Please explain more about the imaging condition that you have used in your data, the problem of low frequency effect, if you have seen in applying RTM on data in the section of the RTM artifacts and explain about the other artifacts that you might have seen in RTM when it is applied on cross hole seismic data.
Please explain about the frequency content of the data that you have acquired during your synthetic data experiments and its effect on detection limit by the RTM method, the computation time and the problem of multiple, if you have seen on your data.
Please explain about the contrast on density and velocity of the boulder in model 2 and explain about the minimum detectable contrast in density and velocity between the boulder and surrounding media, by the proposed method.
Considering the artifacts on the results of RTM on seismic data and the effect of imaging condition on results, I propose to have alook and possibly cite the following publication, which deals with the above mentioned problem:
The first publication is:
Moradpouri, et al. (2016) Seismic Reverse Time Migration Using A New Wave-Field Extrapolator and a New Imaging Condition. Acta Geophys. 64, 1673–1690. https://doi.org/10.1515/acgeo-2016-0076
The second publication is:
Moradpouri, et al. (2017) An improvement in wavefield extrapolation and imaging condition to suppress reverse time migration artifacts. GEOPHYSICS, Volume 82, Issue 6, https://doi.org/10.1190/geo2016-0475.1
What about a model complex model, for instance more than two objects or a faulted object or an object with irregular shape? I see that in model 10, you have modeled a fault with large dip, but what about if faulting, separated the boulder in two parts? Maybe it is more complex, but can better shows the capabilities of the proposed method.
What about using synthetic data, contaminated with random noise? When it is about in version, whether using RTM or FWI method, noise contamination in seismic data is a major problem. Therefore I propose to add a data contaminated with random noise.
What about using real field data example? I know that it requires large computation time, but generally, when a new methodology is propose, it worth to be applied on real field data, even in small size.
Please define the end product in table 1. Also define the inputs and the parameters that have not be described in the text, like as p and r.
The paper holds considerable appeal for practicing exploration seismologist and data scientist in the field of earth science, as it provides valuable insights and practical implications of RTM in inversion of seismic data that are directly applicable to their professional endeavors. The authors added a new methodology in cross hole seismic data processing, making the content highly relevant and engaging for individuals actively involved in oil and gas industry.
The abstract is exceptionally well-crafted, offering a clear and concise overview of the paper's content. It effectively highlights both the specific application of the RTM on cross hole seismic data, and the generic aspects of the work.
The Introduction could benefit from further clarification regarding the application area. While it touches upon the broader context of the research, citing to the neighboring methods that used inversion or RTM in processing of cross hole seismic data, would enhance the reader's understanding from the outset. It is recommended that the authors provide additional information or context in this section to ensure a more seamless transition into the subsequent sections of the paper. This adjustment would contribute to a more comprehensive introduction and improve the overall coherence of the manuscript.
While the Conclusion does touch upon the discussed technologies/methodologies, there is room for improvement in explicitly stating the practical industrial benefits. A more focused articulation of the concrete advantages compared to the competitive methods in the conclusion would enhance the overall clarity and impact of the paper.
Best Regard
Reviewer 2 Report
Comments and Suggestions for Authors
This paper proposed a 3D cross-hole acoustic wave reverse time migration (RTM) image method, validated the correctness of the model, and analyzed the adaptability of various models, with strong engineering significance. However, before acceptance, some modifications are recommended:
1. Improve the format of formulae in the text to align them with the text.
2. In section 3.1, it is mentioned that the grid spacing of the simulation and imaging is dx=dy=dz=0.5m. Would this grid spacing have an impact on the deviation of results?
3. When discussing the influence of sensor settings and observation systems on 3D RTM imaging results by designing various wells and sensor configurations, is it necessary to consider a random distribution, not just the symmetric distribution presented in the paper?
4. Address formatting issues in some titles in the text, such as changing "5.1model-1" to "5.1 Model-1."
5. In the introduction section, it is suggested to split sentences that reference more than four references to enhance clarity in citation expression, such as references [16-19] and [34-37].
6. Please review some possible seismic engineering applications, such as in civil engineering, for example, DOI: 10.1016/j.optlastec.2023.110237
7. Refining this paragraph as suggested will improve the readability and accuracy of the paper.
Comments on the Quality of English LanguagePlease double check english expressions by using professional english language services.
